# Eicosapentaenoic Acid (EPA) Modulates Glucose Metabolism by Targeting AMP-Activated Protein Kinase (AMPK) Pathway

**DOI:** 10.3390/ijms20194751

**Published:** 2019-09-25

**Authors:** Nami Kim, Mi Sun Kang, Miso Nam, Shin Ae Kim, Geum-Sook Hwang, Hyeon Soo Kim

**Affiliations:** 1Western Seoul Center, Korea Basic Science Institute, Seoul 120-140, Korea; 2Department of Anatomy, Korea University College of Medicine, Seoul 136-701, Korea; 3Chemistry & Nanoscience, Ewha Womans University, Seoul 120-750, Korea

**Keywords:** AMPK, EPA, GLUT4, oxygen consumption

## Abstract

EPA, an omega-3 polyunsaturated fatty acid, exerts beneficial effects on human health. However, the molecular mechanisms underlying EPA function are poorly understood. The object was to illuminate molecular mechanism underlying EPA’s role. Here, ^1^H-NMR-based metabolic analysis showed enhanced branched-chain amino acids (BCAAs) and lactate following EPA treatment in skeletal muscle cells. EPA regulated mitochondrial oxygen consumption rate. Furthermore, EPA induced calcium/calmodulin-dependent protein kinase kinase (CaMKK) through the generation of intracellular calcium. This induced the phosphorylation of AMP-activated protein kinase (AMPK) and p38 mitogen-activated protein kinase (p38 MAPK) that led to glucose uptake, and the translocation of glucose transporter type 4 (GLUT4) in muscles. In conclusion, EPA exerts benign effects on glucose through the activation of AMPK-p38 MAPK signaling pathways in skeletal muscles.

## 1. Introduction

Metabolic disorders have become important health problems, because they are associated with numerous diseases including type 2 diabetes, hypertension, atherosclerosis, and ischemic heart disease [1,2]. Although impaired insulin action is a major risk factor for type 2 diabetes [3], the molecular mechanisms underlying this disorder remain unclear. Exercise and fuel deprivation activate AMP-activated protein kinase (AMPK), which leads to the improvement of both insulin resistance and metabolic syndrome in skeletal muscles [4]. AMPK is an energy sensor that regulates cellular and whole-body energy balance and affects the switch from adenosine triphosphate (ATP)-consuming anabolic pathways to ATP-generating catabolic pathways [5]. AMPK exists as a heterotrimeric complex that is composed of a catalytic α subunit, and the regulatory β and γ subunits. Various types of cellular stress lead to a rise in AMP, and a decrease in ATP. As a result, the increased AMP:ATP cellular ratio promotes the activation of AMPK through the phosphorylation of Thr172, which is a critical residue in the kinase domain, located in the activation loop of the α subunit [6,7]. Furthermore, AMPK controls glucose homeostasis by regulating cellular glucose uptake through activating translocation of glucose transporter type 4 (GLUT4) to the plasma membrane [8,9].

Branched-chain amino acids (BCAAs)—such as leucine, isoleucine, and valine—are among the nine essential amino acids in humans, which have crucial roles in physiological regulation [10]. BCAAs have been associated with type 2 diabetes and insulin resistance through the regulation of body weight and glucose homeostasis in human and rodent models [11]. BCAAs reduce plasma glucose levels by stimulating glucose uptake, while improving insulin sensitivity. Since BCAAs affect glucose metabolism, they are hypothesized to be involved with the cellular energy sensor, AMPK.

Eicosapentaenoic acid (20:5n-3, EPA) is an omega-3 polyunsaturated fatty acid, which is found in fish oil and is associated with numerous beneficial health effects [12,13]. EPA has been reported to improve the negative effects of inflammation, obesity, insulin resistance, and type 2 diabetes in several tissues including the liver and skeletal muscles [14]. In addition, EPA promotes insulin sensitivity through the modification of GLUT4, and by increasing the ability of adipocytes to produce adiponectin [15]. Furthermore, EPA prevents coronary artery disease in hypercholesterolemic patients, and decreases Alzheimer’s disease through a neuroprotective effect against excitotoxicity [16,17]. Several studies have reported that EPA activates AMPK to prevent vascular endothelial function in vivo [12]. Moreover, EPA stimulates the activation of AMPK in skeletal muscle cells [18]. This evidence led us to hypothesize that EPA may have a metabolic role via the AMPK-related pathway.

In the present study, we assessed whether EPA exerted a positive effect in the treatment of metabolic diseases, such as diabetes. Our data show that EPA mediates glucose uptake and GLUT4 translocation. Of note, our metabolic analysis using ^1^H-NMR, demonstrated that EPA induces numerous changes in metabolites, especially in BCAAs and -lactate, which induce elevation of the AMP:ATP ratio, and stimulate the AMPK signaling pathway.

## 2. Results

### 2.1. EPA Inhibits Mitochondrial Oxygen Consumption Rate (OCR) and Decreases Intracellular AMP:ATP Ratio in Skeletal Muscle Cells

^1^H-NMR-based metabolic profiling was performed in C2C12 cells to identify whether EPA exerted metabolic effects. Representative 800-MHz ^1^H-NMR spectra were considerably different in control (Appendix A) versus EPA-treated cells (Appendix A). The concentrations of 40 metabolites were measured in cell extracts from five control and EPA-treated samples (Appendix A). We quantified sample differences using principal component analysis (PCA) in C2C12 cells. R^2^X represents the quality of the fit, and Q^2^X indicates the predictability of the PCA model. The PCA score plot showed an explicit separation in the presence and absence of EPA, which was indicated by R^2^X and Q^2^X values of 0.701 and 0.463, respectively (Figure 1A). EPA-stimulated cell extracts showed a considerable increase in the level of BCAAs, compared to controls (Figure 1B). Moreover, EPA significantly increased lactated levels (Figure 1B) [19]. These results indicated that EPA effected metabolic changes in C2C12 myoblasts, especially through the activation of BCAAs, and lactate production.

To investigate mitochondrial respiration, we performed extracellular metabolic flux analysis in C2C12 cells. Mitochondrial oxygen consumption rates (OCRs) were significantly decreased after treatment with EPA and metformin in a dose-dependent manner (Figure 1C). Metformin is known to act on mitochondria and inhibit respiration through complex I (NADH dehydrogenase) inhibition [20,21]. Furthermore, we analyzed mitochondrial stress caused by metabolic changes in the presence of EPA using an XF analyzer. Mitochondrial metabolic extracellular flux analysis showed that EPA significantly decreased basal OCRs compared to controls (Figure 1D). Treatment with carbonilcyanide p-triflouromethoxyphenylhydrazone (FCCP), a mitochondrial membrane uncoupler, slightly increased OCRs in EPA-treated cells, compared to controls. The XF analysis results present three different mitochondrial parameters: basal respiration, proton leak, and ATP production. All metabolic parameters were clearly downregulated in EPA-treated C2C12 cells (Figure 1E). To determine the involvement of adenosine phosphates—such as ATP, ADP, and AMP—we quantified ^1^H-NMR spectra in control and EPA-treated C2C12 cells (Appendix A). The presence of EPA substantially elevated AMP levels, while ATP levels were decreased (Figure 1F). As a consequence, the AMP:ATP ratios were noticeably increased by EPA in C2C12 cells (Figure 1G). These results suggested that EPA regulated mitochondrial respiration by modifying the AMP:ATP ratio in C2C12 cells.

### 2.2. EPA Stimulates Glucose Uptake through the AMPK Signaling Pathway in C2C12 Skeletal Muscle Cells

Exercise and contraction of skeletal muscles rapidly consumes ATP, which leads to an increase in the AMP:ATP ratio, which is responsible for the activation of AMPK [7,22]. We showed that EPA increased the AMP:ATP ratio in C2C12 cells by using ^1^H-NMR metabolic profiling. To examine the metabolic mechanism of EPA in C2C12 cells, we examined the AMPK signaling pathway, which is a key controller of energy metabolism, and regulates glucose uptake. EPA treatment increased the phosphorylation of AMPKα, and of its down-stream target ACC, in a dose- and time-dependent manner in C2C12 cells (Figure 2A,B). The phosphorylation of AMPKα was high at a dose of 50 μM EPA and reached a maximum at 3 h. EPA upregulated glucose uptake in differentiated myotubes in a time-dependent manner (Figure 2C). Furthermore, GLUT4 translocation was increased by EPA treatment, similar to the effects of insulin (Figure 2D). However, these effects were blocked by treatment with an AMPK inhibitor (compound C, Figure 2E,F). GLUT4 translocation to the plasma membrane was elevated in the presence of EPA, as shown by immunocytochemistry (Figure 2G). Together, our results indicated that EPA treatment increased glucose uptake and GLUT4 translocation through the activation of AMPKα in C2C12 cells.

### 2.3. Intracellular Calcium Plays an Upstream Role of AMPK in EPA-Mediated Glucose Uptake in Skeletal Muscle Cells

Because increased cellular calcium levels stimulate the phosphorylation of AMPK through a calcium/calmodulin dependent protein kinase kinase (CaMKK) [23], we hypothesized that intracellular calcium acts upstream of AMPK. Intracellular calcium concentrations were measured in real-time using fluo-3, AM, a calcium indicator, with the aim of determining the role of EPA in C2C12 cells. Treatment with EPA considerably increased green fluorescence intensity in C2C12 cells, an indicator of calcium concentration (Figure 3A). Next, to investigate the specific mechanism of EPA in C2C12 cells, we pre-treated cells with STO-609, a CaMKK inhibitor, and then treated them with EPA. STO-609 treatment abrogated EPA-induced glucose uptake (Figure 3B) and GLUT4 translocation (Figure 3C). The intracellular calcium chelator BAPTA-AM also blocked glucose uptake by EPA (Figure 3D). Further, STO-609 attenuated EPA-activated phosphorylation of AMPK (Figure 3E). Consequently, EPA activated glucose uptake and GLUT4 translocation through the CaMKK-AMPK signaling pathway, mediated by intracellular calcium.

### 2.4. I p38 MAPK Plays a Downstream Role of AMPK in EPA-Mediated Glucose Uptake in C2C12 Skeletal Muscle Cells

GLUT4 activation and glucose uptake is mediated by the p38 MAPK pathway [8,24]. Therefore, to identify the signaling pathway involved in EPA-activated glucose uptake, we examined EPA-mediated activation of p38 MAPK in C2C12 cells. EPA increased the phosphorylation of p38 MAPK in a dose- and time-dependent manner in C2C12 cells (Figure 4A,B). In contrast, pre-treatment with compound C suppressed EPA-induced phosphorylation of p38 MAPK (Figure 4C), which was down regulated by the siRNA-mediated knockdown of AMPKα2 (Figure 4D) and was blocked by a dominant negative mutant of AMPKα2 (K45R); K45R is a kinase-dead AMPKα2 mutant (Figure 4E). Furthermore, p38 MAPK inhibition by SB203580, a p38 MAPK inhibitor, significantly blocked EPA-activated glucose uptake (Figure 4F). These results suggest that EPA-induced glucose uptake is mediated through the p38 MAPK-AMPK signaling pathway.

### 2.5. AS160 is Involved in EPA-Induced GLUT4 Expression in C2C12 Skeletal Muscle Cells

We investigated *GLUT4* expression in C2C12 cells to further examine the EPA-induced upregulation of glucose uptake. GLUT4 is a glucose transporter which is expressed primarily in muscle and fat cells, and which plays a crucial role in whole-body glucose homeostasis [25,26]. EPA-activated C2C12 cells showed an increase in the relative GLUT4 mRNA levels in a time-dependent manner (Figure 5A). In addition, EPA treatment also stimulated GLUT4 protein expression in a time-dependent manner in C2C12 cells (Figure 5B). However, the knockdown of AMPKα2 using an AMPKα2 siRNA blocked EPA-induced GLUT4 protein expression (Figure 5C). GLUT4 translocation is regulated by AS160 [27]. To understand the mechanism underlying EPA-mediated glucose uptake, we examined the effect of EPA on AS160 phosphorylation. EPA-treated C2C12 cells revealed an increase in the phosphorylation of AS160, but this effect was abrogated by knockdown of AMPKα2 (Figure 5D). These results indicate that EPA increased glucose uptake through upregulating GLUT4 expression, via the AS160-AMPK signaling pathway in C2C12 cells.

### 2.6. EPA Activates AMPK and Stimulates Glucose Uptake in Primary Cultured Myoblasts

To investigate the effect of EPA in mice, we then examined the effect of EPA on primary cultured myoblasts. Similar to our previous results, EPA-treated primary cultured myoblasts showed increased phosphorylation of AMPKα and ACC (Figure 6A). To understand the functional importance of AMPK, we performed glucose uptake experiments in myotubes obtained from primary myoblasts. EPA-treated primary cultured myoblasts also revealed an increase in glucose uptake maximally about 1–10 μM ranges (Figure 6B). Moreover, we measured the concentration of intracellular calcium in real-time to determine the mechanism of EPA in primary cultured myoblasts. EPA highly activated the generation of intracellular calcium (Figure 6C). However, compound C and STO-609 blocked EPA-induced glucose uptake effects in primary cultured myotubes, respectively (Figure 6D). As a result, we concluded that EPA activated glucose uptake via the calcium-mediated AMPK signaling pathway in primary cultured myoblasts.

## 3. Discussion

In this study, we investigated how EPA regulates metabolic pathways, and showed that it controls mitochondrial respiration through activation of the AMPK signaling pathway in skeletal muscle cells. Thus, we first confirmed that EPA treatment induced metabolite changes in muscle cells using ^1^H-NMR (Figure 1). Statistical analyses of 40 metabolites comparison using ^1^H-NMR spectra showed a clear separation between controls and EPA treatment in muscle cells (Figure 1A). Moreover, we confirmed that EPA stimulated the activation of BCAAs—such as leucine, isoleucine, and valine—in skeletal muscle cells. BCAAs have recently been recognized as playing a role in insulin sensitivity via the modulation of intracellular AMP or AMP:ATP ratio [11]. EPA improves insulin sensitivity and blood sugar in patients with type 2 diabetes [28]. In addition, we quantified lactate levels following EPA treatment. Several studies have reported that muscle contraction during chronically elevated muscle activity stimulates either glucose to lactate conversion in skeletal muscles [29]. Increased lactate levels are also associated with insulin resistance and type 2 diabetes in adipocytes [30]. We next investigated mitochondrial activity by measuring OCRs to identify the metabolic effect of EPA (Figure 1). Metformin is widely used in type 2 diabetes therapy, and acts as an inhibitor of mitochondrial complex I that activates AMPK [31]. We observed that EPA, similar to metformin, considerably reduced mitochondrial basal OCR level in skeletal muscle cells. EPA also downregulated the proton leak and ATP production compared to control cells. Furthermore, EPA increased the AMP:ATP ratio, which stimulates the phosphorylation of AMPK and activates glucose uptake and the translocation of GLUT4 (Figure 2). Based on these findings, EPA might be correlated with intracellular BCAAs and lactate concentrations through regulation of mitochondrial activity by decreasing glucose through glucose uptake, and by activating the AMPK signaling pathway in muscle cells.

AMPK is an energy status sensor that maintains cellular homeostasis thorough regulation of glucose uptake, glycogen synthesis, mitochondrial biogenesis, insulin sensitivity, and lipid metabolism, including the uptake and oxidation of fatty acids during rest, or during or following exercise [32,33]. Herein, we demonstrated that EPA regulates the calcium signaling pathway, which controls intracellular glucose metabolism through the AMPK-p38 MAPK signaling pathway in muscle cells (Figure 3 and Figure 4). Furthermore, we showed that EPA activates AS160 in muscle cells, but these effects were abolished by the knockdown of AMPKα2 (Figure 5), since AS160 regulates GLUT4 translocation and stimulates contraction-induced glucose uptake [33]. Collectively, these findings suggest that EPA generates intracellular calcium that activates glucose uptake via GLUT4 through the AMPK and p38 MAPK signaling pathways in muscle cells.

In conclusion, the present study suggests that EPA mediates metabolic changes and regulates mitochondrial respiration in skeletal muscle cells. These changes lead to glucose uptake and the lowering of cellular glucose concentration through the stimulation of the AMPK-p38 MAPK signaling pathway and calcium-induced CaMKK activation in skeletal muscle cells (Figure 7). Consequently, we demonstrated that EPA might have the potential to represent a new therapeutic drug for the treatment of metabolic diseases, including type 2 diabetes, obesity, and insulin resistance.

## 4. Materials and Methods

### 4.1. Reagents

cis-5,8,11,14,17-Eicosapentaenoic acid (EPA), metformin, compound C (AMPK inhibitor), insulin, STO-609 (CaMKK inhibitor), SB203580 (p38 MAPK inhibitor), fluo-3, AM and a monoclonal anti-β-actin antibody were purchased from Sigma Chemical Company (St. Louis, MO, USA). BAPTA-AM (intracellular calcium chelator), monoclonal antibody against ACC and polyclonal antibodies against GLUT4 were purchased from Abcam (Cambridge, MA, USA). Monoclonal antibodies against phosphorylated AMPKα, phosphorylated p38 MAPK, p38 MAPK, phosphorylated AS160, AS160, and the polyclonal antibody against AMPKα were obtained from Cell Signaling Technology (Danvers, MA, USA). The polyclonal antibody against phosphorylated ACC was provided by Merck (Rahway, NJ, USA). Fetal bovine serum (FBS) and penicillin-streptomycin were acquired from Thermo Fisher Scientific (Foster City, CA, USA). The monoclonal antibody against c-Myc was acquired from Santa Cruz Biotechnology (Dallas, TX, USA).

### 4.2. Cell Culture

Mouse C2C12 myoblasts were maintained in Dulbecco’s Modified Eagle Medium (DMEM) supplemented with 10% heat-inactivated FBS, 100 U/mL penicillin, and 100 μg/mL streptomycin at 37 °C, in a humidified atmosphere of 5% CO_2_. Rat L6 myoblasts were seeded in 12-well plates at a density of 2 × 10^4^ cells/mL for differentiation into myotubes, that were then used in glucose uptake studies. After 24 h (at >80% confluence), the medium was replaced by DMEM containing 2% (*v*/*v*) FBS. Thereafter, the medium was replaced after 2, 4, and 6 days of culture. Experiments were initiated after 7 days, when myotube differentiation was completed.

### 4.3. Primary Myoblasts Preparation and Culture

Primary myoblasts were isolated from the forelimbs and hindlimbs of three- or four-day-old littermates [34]. The muscles were dissected and minced, disaggregated enzymatically in 4 mL PBS containing 1.5 U/mL dispase II and 1.4 U/mL collagenase D, and triturated with a 10 mL pipette every 5 min for 20 min at 37 °C. The cells were filtered through a 70 µm mesh, and centrifuged at 1000× *g* for 5 min. The cell pellet was dissociated in 10 mL F10 medium supplemented with 10 ng/mL basic fibroblast growth factor and 10% cosmic calf serum. Finally, the cells were pre-plated twice on non-collagen coated plates for 1 h to deplete fibroblasts, which generally adhere faster than myoblasts. For differentiation, the primary myoblasts that were obtained, were cultured to 75% confluence in DMEM containing 5% horse serum. 

### 4.4. NMR Analysis and Data Pre-Processing

Polar metabolites were extracted from cells with a solvent composed of methanol, distilled water, and chloroform. ^1^H-NMR spectra were measured using an 800-MHz NMR instrument. A Noesypresat pulse sequence was applied to suppress the residual water signal. For each sample, 256 transients were collected into 64,000 data points using a spectral width of 16393.4 Hz, with a relaxation delay of 4.0 s and an acquisition time of 2.00 s. All NMR spectra were phased and baseline-corrected using the Chenomx NMR suite version 6.0 (Chenomx Inc., Edmonton, AB, Canada). ^1^H-NMR spectra were segmented into 0.005-ppm bins. Spectral data were normalized to the total spectral area. Data files were imported into MATLAB (R2006a; Mathworks, Inc., Natick, MA, USA), and all spectra were aligned using the correlation optimized warping (COW) method [35].

### 4.5. Cellular Metabolic Rate

Cell metabolic profile was measured by using Seahorse XFp and XF24 analyzers (Agilent Technologies, Santa Clara, CA, USA). C2C12 cells were seeded in pre-coated XFp cell culture microplates at 8 × 10^4^ cells/well, and incubated overnight at 37 °C with 5% CO_2_. The cells were treated with 50 μM EPA for 6 h, and the medium was replaced with unbuffered DMEM supplemented with 10 mM glucose, 2 mM l-glutamine, and 1 mM pyruvate (Sigma Chemical, Moscow, Russia). Each cycle included 3 min of mixing, 2 min of incubation, and a 2 min measurement. Three measurements were obtained at baseline, and following an injection of 1 μM oligomycin, 0.5 μM FCCP, and 0.5 μM rotenone/antimycin A. Mitochondrial respiration was quantified based on the oxygen consumption rate.

### 4.6. Western Blot Analysis

The cells were grown in six-well plates. After achieving 80–90% confluence, the cells were serum starved for 3 h before treatment with selected agents. The cells were then treated with 50 μM EPA for 3 h. After treatment, the medium was aspirated, and cells were washed twice with ice-cold phosphate-buffered saline (PBS) and lysed in RIPA buffer (0.5% deoxycholate, 0.1% sodium dodecyl sulfate (SDS), 1% Nonidet P-40, 50 mM NaCl, and 50 mM Tris-HCl) containing protease and phosphatase inhibitor cocktails (Sigma Chemical Company). The supernatants were centrifuged for 20 min and then heated for 15 min at 99 °C. After separation on a 10% SDS-polyacrylamide gel, proteins were transferred onto polyvinylidene difluoride (PVDF) membranes. The membranes were incubated with primary antibodies at 4 °C overnight, after which they were washed three times with Tris-buffered saline (TBS) containing 1% Tween-20. The membranes were then incubated with horseradish peroxidase (HRP)-conjugated secondary antibodies for 1 h. The anti-β-actin antibody was used to normalize protein loading. The blots were visualized using ECL solution (Thermo Fisher Scientific, Waltham, MA, USA). Quantitation was performed by densitometry using ImageJ (National Institutes of Health, Bethesda, MD, USA).

### 4.7. Assessment of Intracellular Calcium

The intracellular calcium concentration was measured by detecting the fluorescence of cells treated with fluo-3, AM, a sensitive calcium indicator. Fluorescence was detected using a Zeiss LSM 700 confocal microscope (Zeiss, Oberkochen, Germany). The cells were treated with 5 μM fluo-3, AM in culture medium for 45 min, in a CO_2_ incubator. After washing with medium, the cells were incubated in the absence of fluo-3, AM for 15 min to allow for the complete de-esterification of the dye. Culture plates were placed on a temperature-controlled microscope stage and were observed under a 20 × objective. The signal was detected at an excitation and emission wavelength of 488 nm.

### 4.8. Uptake of 2-deoxy-d(H^3^)-glucose

Glucose uptake was determined by measuring the uptake of 2-deoxy-d(H^3^)-glucose (2-DG) in differentiated L6 myotubes. The cells were rinsed twice with warm PBS (37 °C) and then starved in serum-free DMEM for 3 h. After EPA treatment, the cells were incubated in KRB (20 mM HEPES (pH 7.4), 130 mM NaCl, 1.4 mM KCl, 1 mM CaCl_2_, 1.2 mM MgSO_4_, and 1.2 mM KH_2_PO_4_) containing 0.5 μCi 2-DG at 37 °C for 15 min. The reaction was terminated by placing the plates on ice, and by washing the cells twice with ice-cold PBS. The cells were then lysed in 0.5 N NaOH, and 400 μL cell lysate was mixed with 3.5 mL scintillation cocktail. Radioactivity was measured using a scintillation counter.

### 4.9. Myc-GLUT4 Translocation Assay

Cell surface expression of Myc-GLUT4 was quantified by performing an antibody-coupled colorimetric absorbance assay, as described previously [36]. After EPA treatment, differentiated L6 myotubes that stably expressed Myc-GLUT4 were incubated with polyclonal anti-c-Myc antibody for 60 min, fixed with 4% paraformaldehyde in PBS for 10 min, and incubated with HRP-conjugated goat anti-rabbit antibody for 1 h. The cells were then washed six times with PBS and incubated in 1 mL *o*-phenylenediamine (0.4 mg/mL) for 30 min. The absorbance of the supernatant was measured at 492 nm.

### 4.10. RT-Qpcr

Total RNA was extracted from 1 × 10^6^ cells/mL using an RNeasy Mini Kit (Qiagen, Valencia, CA, USA) according to the manufacturer’s protocol. RNA concentration and quality were immediately determined using a Nanodrop 2000 (Thermo Fisher Scientific). Fifty ng of RNA were used as a template for cDNA synthesis, using the GoTaq^®^ 1-Step RT-qPCR System, according to the manufacturer’s instructions (Promega, Madison, WI, USA). Reactions were carried out with SYBR green for 40 cycles of denaturation at 95 °C for 10 s, annealing at 60 °C for 30 s, and extension at 72 °C for 30 s using the StepOnePlus Real-Time PCR System (Applied Biosystems, Foster City, CA, USA). The qPCR was performed using specific primers for *GLUT4*, *LCAD*, *CPT1a*, *ACOX*, *FABP4*, *UCP2*, and β-actin. The experiment was performed using three independent biological replicates. Gene expression was normalized to the mRNA expression level of β-actin, as an endogenous control, and fold changes were calculated between the treated and the control samples.

### 4.11. Silencing of Genes Encoding AMPKα2 and p38 MAPK

Cells were seeded in six-well plates, cultured for 24 h to 70% confluence, and then transiently transfected with siRNAs against genes encoding AMPKα2 (L-040809, Dharmacon, GE Healthcare, Little Chalfont, Buckinghamshire, UK) and p38 MAPK (L-040125, Dharmacon) using Lipofectamine 2000 (Invitrogen, Life Technologies, Carlsbad, CA, USA), according to the manufacturer’s protocol. For transfection, the siRNAs and 8 µL Lipofectamine 2000 were diluted using 500 µL reduced serum medium and mixed well. The mixture was incubated for 20 min at room temperature, and then added dropwise to each culture well containing 1.5 mL serum-free medium. The medium was replaced with fresh complete medium after 6 h of transfection.

### 4.12. Cytochemistry

Cells were fixed with 4% PFA/PBS and permeabilized with 0.2% Triton-X 100. After blocking with 0.2% bovine serum albumin for 30 min, cells were incubated with an anti-GLUT4 antibody at 1:500 dilution for 60 min and then probed with a Cy3-labeled secondary antibody (Molecular Probe, Eugene, OR, USA). Stained cells were visualized using a confocal microscope.

### 4.13. Statistical Analysis

Multivariate statistical analyses of ^1^H-NMR data were performed with Pareto scaling using SIMCA-P+ software, version 12.0 (Umetrics, Umeå, Sweden). All changes in metabolite levels—including isoleucine, leucine, valine, lactate, fumarate, malate, AMP, ADP, and ATP levels—were assessed using the Student’s *t*-test with GraphPad Prism (version 5 for Windows; GraphPad Software, La Jolla, CA, USA). Data are presented as mean ± standard deviation (SD) of individual experiments. All experiments were performed with at least three independent replicates. The difference between mean values was considered statistically significant at *p* < 0.05.

## Figures and Tables

**Figure 1 ijms-20-04751-f001:**
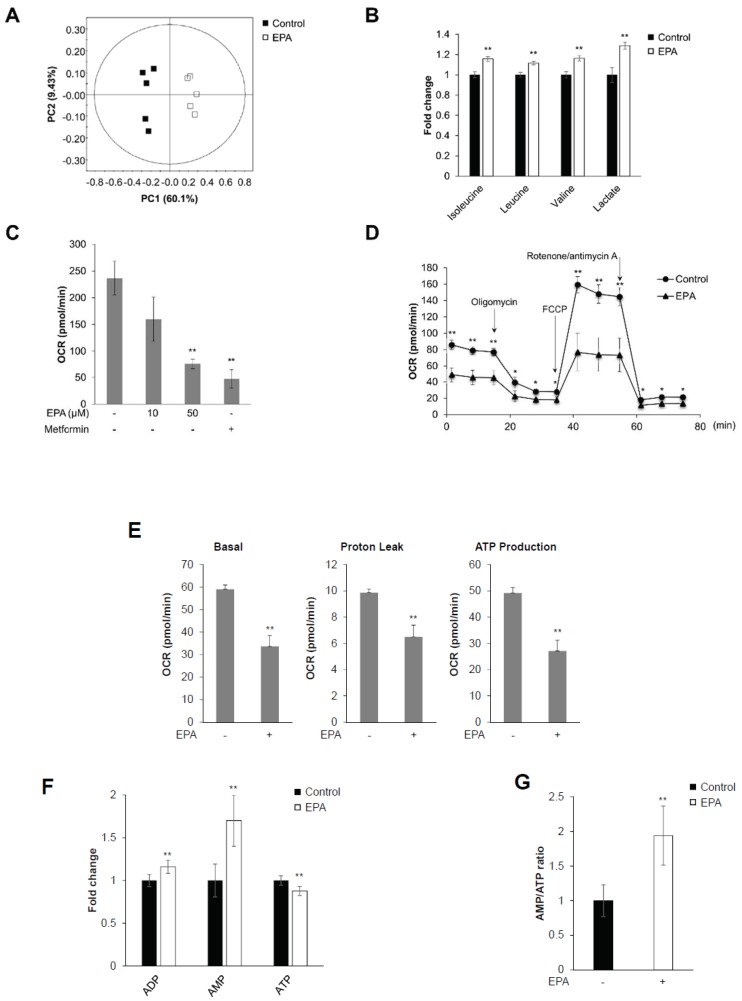
EPA inhibits mitochondrial oxygen consumption rate (OCR) and decreases intracellular AMP:ATP ratio in skeletal muscle cells. C2C12 cells were stimulated with 50 μM EPA for 3 h. Cellular metabolites were extracted with MeOH/water/CHCl_3_, and NMR-based metabolic profiling was conducted. (**A**) PCA score plot for ^1^H-NMR spectra of cellular metabolite extract levels for metabolome analysis. (**B**) Levels of BCAAs (isoleucine, leucine, and valine), lactate, fumarate, and malate in C2C12 cellular metabolite extracts. (**C**) C2C12 cells treated with the indicated dose of EPA and 10 mM metformin for 18 h, respectively. Mitochondrial oxygen consumption rate (OCR) measured using an XF24 analyzer. Metformin was used as a positive control. (**D**) Mitochondrial OCR in EPA (50 μM) stimulated cells measured using an XFp analyzer in response to 1 μM oligomycin, 0.5 μM FCCP, and 0.5 μM rotenone/antimycin A. (**E**) Basal respiration, proton leak, ATP production calculated from (D). (**F**) ^1^H-NMR-based metabolic profiling analysis showing the intensity of the adenosine phosphates ATP, ADP, and AMP in EPA-treated or untreated cellular metabolite extracts. (**G**) AMP:ATP ratio derived from ^1^H-NMR spectral intensities. * *p* < 0.05, ** *p* < 0.01 compared to untreated cells. Results from three independently replicated experiments are presented.

**Figure 2 ijms-20-04751-f002:**
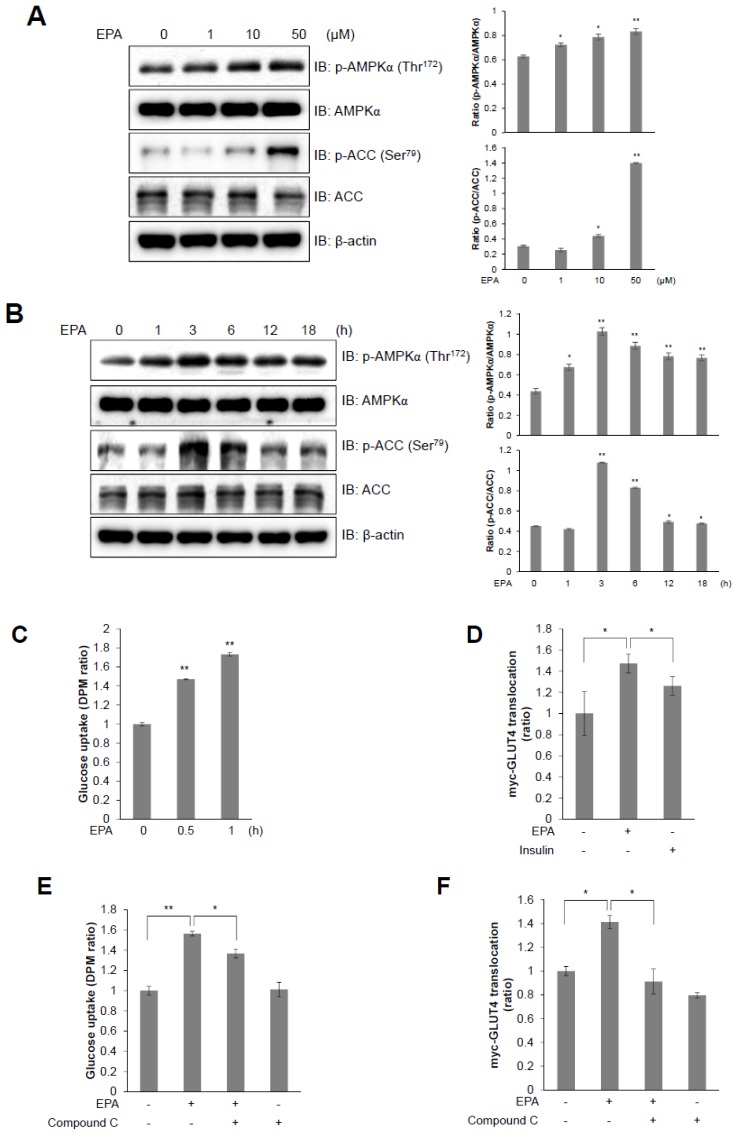
T EPA stimulates glucose uptake through the AMPK signaling pathway in C2C12 myoblasts. (**A**) Western blot analysis of AMPKα and ACC phosphorylation in C2C12 cells treated with various concentrations of EPA for 3 h or (**B**) 30 µM EPA for the indicated times. Total protein levels for AMPKα, ACC and β-actin were used as loading controls. (**C**) 2-deoxy-d[H^3^]-glucose (2-DG) uptake measured in L6 cells differentiated for 7 days and treated with 50 μM EPA for the indicated times. (**D**) Cell surface expression of Myc-GLUT4 quantified using an antibody-coupled colorimetric absorbance assay in myoblasts stably expressing L6-GLUT4-myc, differentiated for 7 days, and treated with EPA or 100 nM insulin for 3 h. (**E**) Differentiated L6 myotubes treated with 50 μM EPA for 3 h in either the presence or absence of compound C (5 μM). (**F**) Differentiated L6-myc-GLUT4 cells were pre-treated with compound C for 30 min, and then incubated with EPA for 3 h. The Myc-GLUT4 expression in cells is quantified using an absorbance assay. (**G**) Representative images (GLUT4, DAPI, and merge) of cells treated with EPA for 3 h. Insulin (100 nM) was used as positive control. Scale bar, 20 μm. * *p* < 0.05, ** *p* < 0.01 compared to untreated cells. Results from three independently replicated experiments are presented.

**Figure 3 ijms-20-04751-f003:**
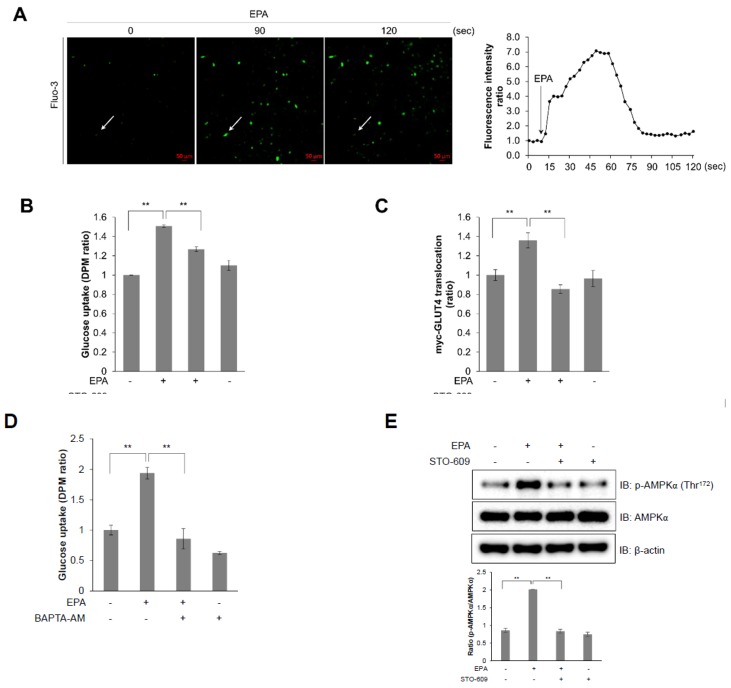
Intracellular calcium plays as an upstream of AMPK in EPA-mediated glucose uptake in skeletal muscle cells. (**A**) C2C12 cells pre-treated with fluo-3 AM for 30 min, and then treated with EPA. (**B**) Uptake of 2-DG in differentiated L6 myotubes pre-treated with STO-609 (5 µM) and incubated with EPA for 3 h. (**C**) Differentiated L6-GLUT4-myc cells were pre-treated with STO-609, and incubated with EPA for 3 h. (**D**) Uptake of 2-DG in differentiated L6 myotubes treated with BAPTA-AM (25 µM), and incubated with EPA for 3 h. (**E**) Western blot analysis of cell lysates from C2C12 cells pre-treated with 5 µM STO-609 for 30 min, and incubated with EPA. ** *p* < 0.01 compared to untreated cells. Results from three independently replicated experiments are presented.

**Figure 4 ijms-20-04751-f004:**
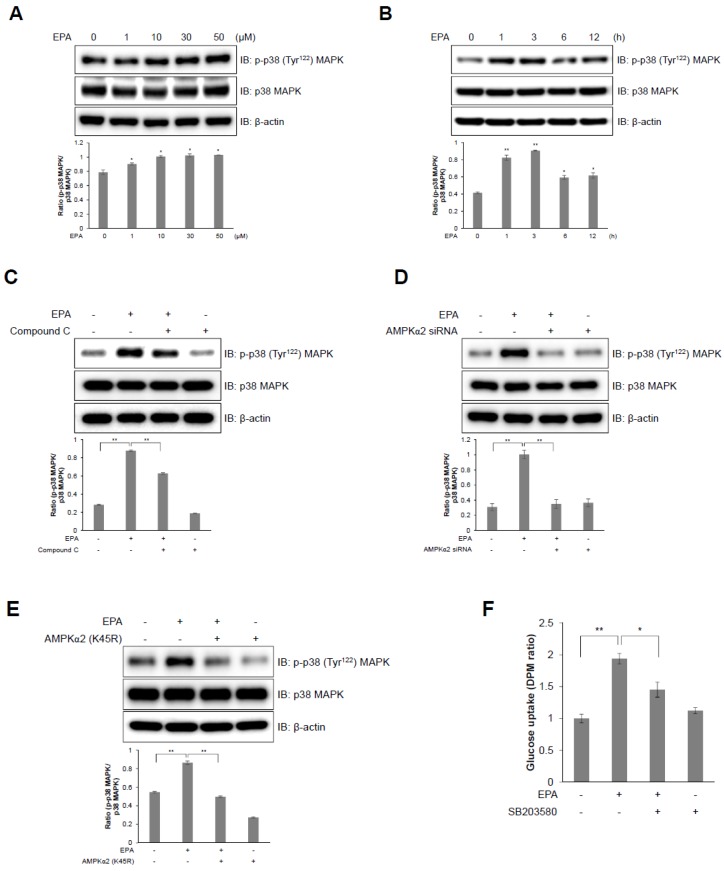
p38 MAPK plays as a downstream of AMPK in EPA-mediated glucose uptake in C2C12 myoblasts. (**A**) Western blot analysis of p38 MAPK phosphorylation in C2C12 cells treated with different concentrations of EPA for 3 h; or (**B**) 50 µM EPA for the indicated times. (**C**) C2C12 myoblasts were treated with 50 µM EPA for 3 h in the presence of 5 µM compound C. The levels of total p38 MAPK and β-actin are presented as loading controls. (**D**) Western blot analysis of cells transfected with AMPKα2 siRNA (100 nM) for 2 days, and then treated with EPA. (**E**) C2C12 cells were transfected with AMPKα2 K45R plasmid (4 µg) and then treated with EPA. Proteins were extracted from the transfected cells and protein levels were measured via western blot analysis. (**F**) Uptake of 2-DG in differentiated L6 myotubes pre-treated with SB203580 (5 µM) and incubated with EPA for 3 h. * *p* < 0.05, ** *p* < 0.01 compared to untreated cells. Results from three independently replicated experiments are presented.

**Figure 5 ijms-20-04751-f005:**
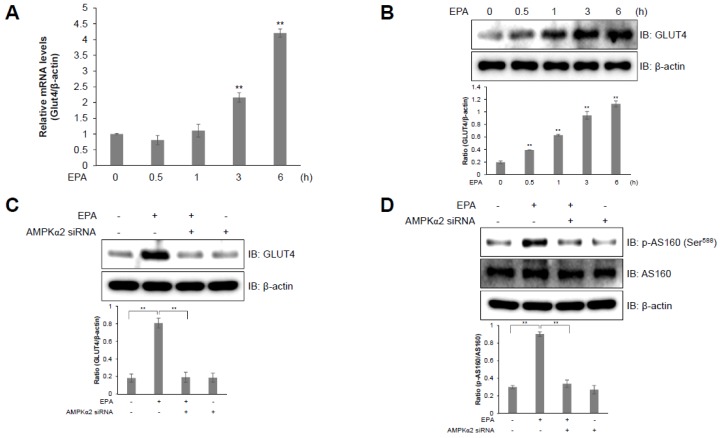
AS160 is involved in EPA-induced GLUT4 expression in C2C12 myoblasts. (**A**) GLUT4 and β-actin transcripts measured by RT-qPCR from EPA-treated C2C12 cells. (**B**) Western blot analysis of GLUT4 and β-actin levels in cells stimulated with 50 μM EPA for the indicated times. The levels of β-actin are presented as a loading control. (**C**) Western blot analysis of cell lysates from cells transfected with AMPKα2 siRNA (100 nM) for 2 days and then treated with EPA. (**D**) p-AS160, AS160, and β-actin expression quantified using western blot analysis in cells transfected with AMPKα2 siRNA and treated with EPA. ** *p* < 0.01 compared to the basal condition. Results from three independently replicated experiments are presented.

**Figure 6 ijms-20-04751-f006:**
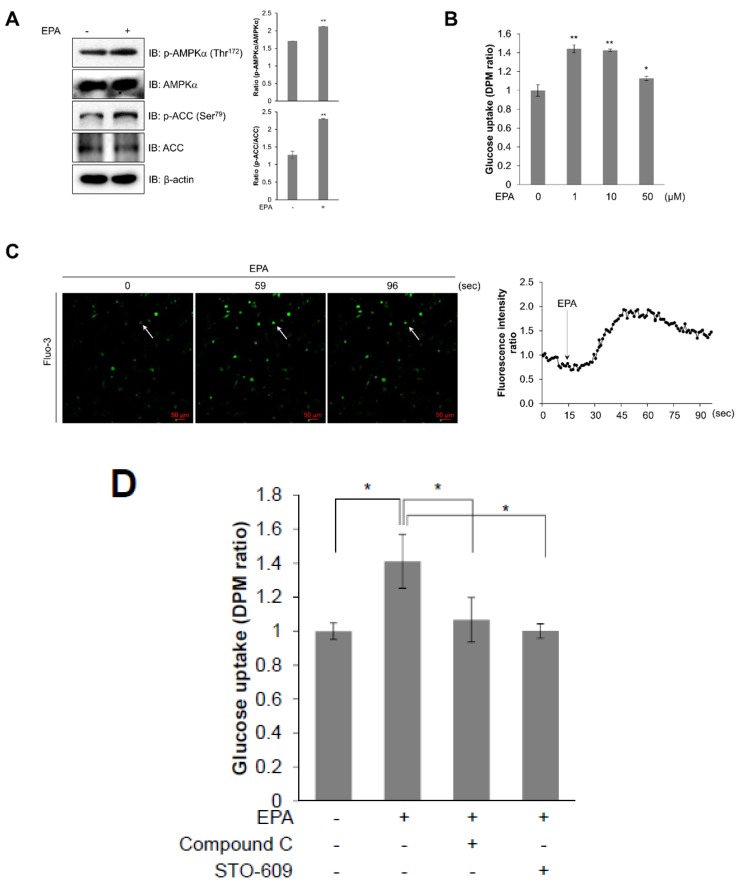
EPA activates AMPK and stimulates glucose uptake in primary cultured myoblasts. (**A**) Western blot analysis of phosphorylated AMPKα and ACC in primary cultured myoblasts stimulated with 50 μM EPA for 3 h. Blotting with antibodies specific to the non-phosphorylated ACC and AMPKα isoforms, and β-actin served as loading controls. (**B**) Primary myoblasts were differentiated for 5 days, and measured glucose uptake after treated with different concentrations of EPA for 3 h. (**C**) Confocal microscopy was used to measure intracellular calcium concentration. Primary-prepared myoblasts were pre-treated with fluo-3 AM for 30 min and then treated with EPA. (**D**) To measure uptake of 2-DG in differentiated primary myotubes, cells were pre-treated with 5 μM of compound C or STO-609 for 30 min and treated with EPA for 3 h. * *p* < 0.05, ** *p* < 0.01 compared to untreated cells. Results are from three independently replicated experiments are presented.

**Figure 7 ijms-20-04751-f007:**
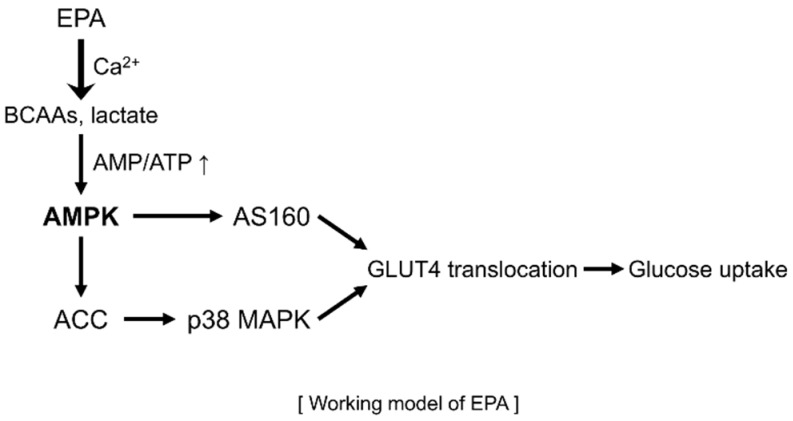
Schematic illustration of AMPK signaling pathway following EPA treatment in skeletal muscle cells.

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
