# Peer review of "Eicosapentaenoic Acid (EPA) Modulates Glucose Metabolism by Targeting AMP-Activated Protein Kinase (AMPK) Pathway"

_ijms, 2019, doi:10.3390/ijms20194751_

Round 1

Reviewer 1 Report

The current manuscript by Kim et al. is focused on the study of the effect of eicosapentaenoic Acid (EPA) on skeletal muscle metabolism. In particular they contribute to the current literature reporting that EPA treatment modulates AMP:ATP ratio thus activating AMPK and regulating glucose uptake. The study is of potential interest as it is intended to clarify the molecular mechanisms involved in EPA effects in skeletal muscle, with potential application in diabetes.

Major concerns:

It is very important that the authors clearly specify in the text if the experiments are performed in myoblasts or in myotubes. Indeed, myotubes represent a model of differentiated skeletal muscle fibers while myoblasts are proliferating cells involved in skeletal muscle repair: the phenotype of the two models is completely different and related to the differentiation process. Moreover, the authors performed the experiments related to the study of EPA-induced modulation of cellular signaling in C2C12 mouse myoblasts while the experiments related to the biological effect of the omega-3 polyunsaturated fatty acid (GLUT4 translocation and glucose uptake) are performed in L6 rat myotubes, why the authors change the cellular model? The use of different cellular model should be clearly stated in the text. Even if the experiments are also performed in primary myoblasts/myotubes it could be of interest to validate the obtained results using only one cellular model for both myoblasts and myotubes. GLUT4 translocation is quantified with a not convincing method, an alternative method (e. g. immunofluorescence, cytofluorimetric analysis or subcellular fractionation) should be used to confirm the obtained results. Moreover, GLUT4 translocation experiments are performed in L6 myotubes overexpressing myc-GLUT4,  a forced system to study GLUT4 response to EPA, what about the translocation of physiological GLUT4 in response to EPA?

Minor concerns

The statement “EPA significantly activates branched-chain amino acids (BCAAs)” (page 1 line 16) is not corrected: since the NMR-metabolomic studies are performed in cell lysates maybe the authors can hypothesize that BCAA uptake is enhanced following EPA treatment? What about BCAA concentration in the medium after EPA treatment? Is it reduced? Page 1 line 16: “EPA significantly activates (branched-chain amino acids (BCAAs)) and TCA cycle intermediates in skeletal muscle cells that correlate with mitochondrial activity”: what the authors mean? The TCA intermediate (fumarate and malate are measured by NMR metabolomic studies) concentration is not affected by EPA treatment while oxygen consumption is reduced (that suggests a glycolytic switch), how the authors explain these results? Lactate is not a metabolite of the TCA cycle, please amend the text (e.g. page line 2 lines 75,76); since fumarate and malate levels are not altered by EPA treatment the TCA cycle intermediates concentration is not affected by EPA treatment, please amend the text (e.g page 2 line 78). Legend to Figure 3 c (page 6 line) is not corrected as in figure is reported the translocation of MYC-GLUT4 and not the 2-DG Page 9 line 216: glucose uptake experiments are performed in myotubes obtained from primary myoblasts and not in “in primary cultured myoblasts” Page 9 line 218 substitute uM with µM

Author Response

Responses to the Reviewers’ Comments

We deeply appreciate the thorough analysis and constructive suggestions on the manuscript by all two reviewers. As requested by the reviewers, we have now addressed each of the issues raised in a satisfactory manner. With these revisions, we believe that we have satisfactorily addressed all the concerns. 

Reviewer 1

Major concerns:

Why the authors change the cellular model? The use of different cellular model should be clearly stated in the text. Even if the experiments are also performed in primary myoblasts/myotubes it could be of interest to validate the obtained results using only one cellular model for both myoblasts and myotubes.

Among skeletal muscle cells, differentiated L6 myotubes showed higher glucose uptake than

C2C12 cells, suggesting that L6 myotubes were the most promising model for investigating glucose uptake.

The reference is that “Glucose uptake in human and animal muscle cells in culture. Biochemistry and Cell Biology (1990), Sarabia V, Ramlal T & Klip A, 68 536-542”

GLUT4 translocation is quantified with a not convincing method, an alternative method (e. g. immunofluorescence, cytofluorimetric analysis or subcellular fractionation) should be used to confirm the obtained results. Moreover, GLUT4 translocation experiments are performed in L6 myotubes overexpressing myc-GLUT4, a forced system to study GLUT4 response to EPA, what about the translocation of physiological GLUT4 in response to EPA?

As suggested, we have added immunohistochemistry data to Figure 2g.

Minor concerns

The statement “EPA significantly activates branched-chain amino acids (BCAAs)” (page 1 line 16) is not corrected: since the NMR-metabolomic studies are performed in cell lysates maybe the authors can hypothesize that BCAA uptake is enhanced following EPA treatment?

What about BCAA concentration in the medium after EPA treatment? Is it reduced? Page 1 line 16: “EPA significantly activates (branched-chain amino acids (BCAAs))

As requested, we have modified to “Here, 1H-NMR-based metabolic analysis showed enhanced branched-chain amino acids (BCAAs) and lactate following EPA treatment in skeletal muscle cells”

We have deleted some sentence related with TCA cycle intermediates.

We couldn’t check the concentration of BCAA in the medium. Because medium contains excessive nutrients (FBS, glucose....), protein and metabolites, it is difficult to extract pure metabolites from medium. There is another way to track uptake using isotopes (e.g C13) to demonstrate that BCAA is directly uptake to intracellular, but so far, the conditions have not been established in our lab. However, our lab has developed how to extract metabolites from skeletal muscle cells and analyze global metabolites through NMR. As a result, it was confirmed that BCAA is significantly changed by EPA.

TCA cycle intermediates in skeletal muscle cells that correlate with mitochondrial activity”: what the authors mean? The TCA intermediate (fumarate and malate are measured by NMR metabolomic studies) concentration is not affected by EPA treatment while oxygen consumption is reduced (that suggests a glycolytic switch), how the authors explain these results? Lactate is not a metabolite of the TCA cycle, please amend the text (e.g. page line 2 lines 75,76); since fumarate and malate levels are not altered by EPA treatment the TCA cycle intermediates concentration is not affected by EPA treatment, please amend the text (e.g page 2 line 78).

We have deleted several sentences related to TCA cycle intermediates.

As suggested, we have modified sentence to “Moreover, EPA significantly increased lactated levels (Figure 1b)”

We also revised Figure 1b, which relates to TCA cycle intermediates.

Legend to Figure 3 c (page 6 line) is not corrected as in figure is reported the translocation of MYC-GLUT4 and not the 2-DG

As requested, we have modified to “Differentiated L6-GLUT4-myc cells were pre-treated with STO-609, and incubated with EPA for 3 h”

Page 9 line 216: glucose uptake experiments are performed in myotubes obtained from primary myoblasts and not in “in primary cultured myoblasts”

We have changed.

Page 9 line 218 substitute uM with µM

We have corrected to “µM”

Reviewer 2 Report

This study of the molecular mechanisms of EPA's health benefits on metabolic disease is important and interesting to the general public.The manuscript is well organized and presented. The data are sufficient to  support the conclusion.

Some minor changes are recommended as follow

1. The discussion can be expanded to further explain that  EPA EPA's molecular mechanism. If the author can add a graph that summarizes the EPA 's influence in the cellular pathway in the discussion session, the reader can have a better understanding of this study.

2.  In the discussion session, the figure number should be referred when the results are discussed. (For example, in line 240, 252, 257, when the results were mentioned, it is not clear where to find them)

3.  Some language used in the discussion session is not very clear. ( For example, line 240, ...showed a clear separation between controls and EPA treatment..... is this the data shown in Figure 1 A ? Further explanation will be helpful.

4.  Some icons in the graphs are hard to read. Suggest to enlarge some of the figures.

Author Response

Reviewer 2

Minor concerns

The discussion can be expanded to further explain that EPA EPA's molecular mechanism. If the author can add a graph that summarizes the EPA 's influence in the cellular pathway in the discussion session, the reader can have a better understanding of this study.

As suggested, we have added schematic illustration to Figure 7.

In the discussion session, the figure number should be referred when the results are discussed. (For example, in line 240, 252, 257, when the results were mentioned, it is not clear where to find them)

As requested, we have added figure number from Figure 1 to 5.

Some language used in the discussion session is not very clear. (For example, line 240, ...showed a clear separation between controls and EPA treatment..... is this the data shown in Figure 1 A ? Further explanation will be helpful.

As you suggested, we have modified to “Statistical analyses of 40 metabolites comparison using 1H-NMR spectra showed a clear separation between controls and EPA treatment in muscle cells (Figure 1a).

Some icons in the graphs are hard to read. Suggest to enlarge some of the figures.

As suggested, we have now modified some figures. Figure 1D, Figure 3A and Figure 6C have been modified.

Round 2

Reviewer 1 Report

I am satisfied with the changes made to the manuscript and with the answers of the authors.